# Changes to Secondary School Physical Activity Programs and Policy after Emerging from COVID-19 Lockdowns

**DOI:** 10.3390/ijerph21060788

**Published:** 2024-06-17

**Authors:** Hussain Chattha, Markus J. Duncan, Negin A. Riazi, Scott T. Leatherdale, Karen A. Patte

**Affiliations:** 1Department of Health Sciences, Brock University, 1812 Sir Isaac Brock Way, Saint Catharines, ON L2S 3A1, Canada; 2Healthy Active Living and Obesity Research Group, Children’s Hospital of Eastern Ontario Research Institute, 401 Smyth Road, Ottawa, ON K1H 8L1, Canada; 3Student Health and Wellbeing, University of British Columbia, 2329 West Mall, Vancouver, BC V6T 1Z4, Canada; 4School of Public Health Sciences, University of Waterloo, 200 University Ave West, Waterloo, ON N2L 3G1, Canada

**Keywords:** adolescents, physical education, urbanicity, physical activity promotion, sports, fitness

## Abstract

The purpose of this study was to explore the adaptations that schools made to physical activity programs and facilities, and disparities by area urbanicity and income, during the first school year after the emergence of the COVID-19 pandemic. In a convenience sample of 132 secondary schools in Canada, school contacts responded to an annual survey in the 2020–2021 school year on changes to physical activity programs and facilities, and related staff training. Content analysis categorized open-ended text responses, and schools were compared based on area urbanicity and median income. Most schools canceled all interschool sports (88.9%) and intramurals (65.9%). New programs were added by 12.6% of schools, and about half (49%) of schools reported some continuing programs, most of which were sports programs, followed by facility and equipment access. Physical activity facilities were closed in 18.1% of schools, while 15.7% had new facilities added, and 11% temporarily converted facilities into learning spaces. Large/medium urban schools were at greater odds of having made any change to their facilities compared to schools in rural/small urban areas (odds ratio (95% confidence interval): 2.3 (1.1, 4.8)). The results demonstrate the considerable scale and nature of the restrictions in school provisions of physical activity opportunities during this period, as well as the resourcefulness of some schools in adding new programs and facilities.

## 1. Introduction

The response to the coronavirus disease 2019 (COVID-19) pandemic caused numerous and varied changes to the normal operation of secondary schools across Canada. Measures to reduce the spread of disease, such as school closures and transitions to online learning, contributed to wholesale changes in adolescent health behaviors, including significant reductions in physical activity (PA) [1,2,3]. Emerging research additionally highlights significant decreases in organized PA (e.g., team sports) and reduced opportunities for unorganized PA, due to closures of schools, playgrounds, parks, and recreational and sports facilities [4,5]. The benefits of PA in all aspects of health are well documented. 

The value of face-to-face learning has become more evident due to the fallout from the pandemic. An in-person school environment enables a wealth of opportunities for students that effectively promote PA, including access to sports, clubs, teams, and resources (e.g., equipment, spaces, facilities, partners, and coaches) [6]. Schools also offer important opportunities for children to acquire the necessary skills and knowledge that underlie PA [7]. Research suggests that school-based PA programs may increase the proportion of students participating in PA, the duration of time spent in PA, and overall fitness levels [8]. Furthermore, school may be one of the only options for some students to access organized and safe options for PA [9]; they offer an ideal context for equitable PA promotion as the large majority of adolescents attend school and spend most of their weekdays there. Online PA programs may not be as attractive to some individuals due to the lack of social interaction and guidance; however, when in-person opportunities are limited, they may serve as a lower-risk and accessible temporary substitute to regular PA facilities [10]. A U.S.-based study using early pandemic data suggested that online-based PA programs may be better received by students in the future years; one-third of the sample had taken up newly introduced remote or streaming team sports or PA classes due to in-person restrictions [11]. 

Canadian secondary schools’ responses to the pandemic and their containment measures varied by jurisdiction. Secondary schools in Ontario and Alberta experienced the longest suspensions of face-to-face instruction at 19 weeks from 14 March 2020 to 15 May 2021, while in British Columbia and Quebec, closures lasted 9 and 10 weeks, respectively [12]. Schools in lower-income neighborhoods faced prolonged closures and shifts to online learning, and adolescents with lower socioeconomic or racialized backgrounds were particularly affected by these school closures [12]. Quebec schools that had a socioeconomic disadvantage due to having fewer resources were less likely than advantaged schools to have offered extracurricular activities, including competitive sports, physical activities, free gym access, and special interest clubs [13]. How schools have adapted their PA programs to the first full year of instruction during the COVID-19 pandemic is not well known. While extant literature largely focused on closures of facilities and their impact on students, little research has explored what new programs, facilities, and staff training schools have introduced to strike a balance between reducing the spread of COVID-19 and adequately promoting PA. The COVID-19 pandemic required considerable adjustment for schools. Remote instruction was new for most educators and school staff, offering effective PA instruction and programs online may require additional training. School staff training in physical education is important in the successful implementation of programs that lead to better PA outcomes for students [14,15]. The types of training received by staff, and whether staff are more likely to be trained depending on the school’s regional median income or urbanicity, also remain unexplored. Additionally, research has primarily focused on elementary school-level students, and little attention has been given to how secondary school students have been impacted and restricted from participating in PA throughout the pandemic. 

As key context for the promotion of PA among young people, further understanding of how the school PA context changed during a period of disruption may provide valuable insight to better prepare for future events. Beyond understanding the scale and nature of the reduced ability of schools to provide PA opportunities, schools may have been resourceful in offering new and adapted PA programs, including during online learning, that could point to novel avenues for promoting PA. Fresh strategies targeting this critical developmental stage are needed, as PA typically declines over adolescence, and many lifelong lifestyle habits become established [16]. Furthermore, and not specific to the pandemic period, there has long been a limited number of large-scale studies published with comprehensive assessments of school PA environments that are able to assess disparities. iThe current study aimed to understand how Canadian secondary schools adapted to the pandemic by exploring what PA programs and facilities schools have newly introduced, removed, or changed, and what PA training school staff received during the first full year of instruction following the onset of the COVID-19 pandemic. A secondary aim was to examine whether changes in school PA program, facility, and staff training varied by school area urbanicity and regional median household income.

## 2. Methods

### 2.1. Study Design and Participants

The COMPASS study is an ongoing prospective cohort study that collects annual survey data from a convenience sample of Canadian secondary schools and students [17]. Whole school and school board samples were selected and enrolled based on the permitted use of active-information passive-consent protocols. The current study used school-level data pertaining to changes in PA programming during the COVID-19 pandemic that were collected via the School Programs and Policies Questionnaire (SPP) [18]. The SPP is an online survey completed annually by the administrators and/or staff at each school that are most knowledgeable about the school health environment. School contacts are encouraged to complete the survey as a team and are provided their survey responses from the former year of participation to assist with recall. The SPP was developed to be a brief yet comprehensive assessment of the school policy and program environment, via modifying the Healthy School Planner tool [19], with the goals to minimize school burden while maximizing psychometric properties. COMPASS school-level measures were developed with input from school and public health stakeholders and knowledge holders, and researchers from diverse disciplines (with expertise in school and youth health, physical activity, survey development, school policy, etc.). Data obtained from previous years have informed revisions to continually improve the SPP measures (e.g., based on missing data, or any ambiguous or vague responses) [18]. The survey included both qualitative and quantitative items to collect school-level data on PA programs, policies, and resources. A COMPASS staff member followed up with school contacts by phone and/or e-mail in attempt to clarify any missing or unclear survey responses.

This study used cross-sectional data from 132 schools across Alberta, British Columbia, Ontario, and Quebec from the ninth year of the COMPASS study (2020–2021 school year). Complete details on COMPASS study protocols are available in print [17] and online (https://uwaterloo.ca/compass-system/).

### 2.2. Measures

#### 2.2.1. School Descriptors

School-level data included urbanicity, regional median income, and school delivery mode. School area urbanicity was classified based on Statistics Canada definitions, including “large urban”, “medium urban”, “small urban, and “rural” [20]. Urbanicity groups were dichotomized as “large urban” and “medium urban” in one group, and “small urban” and “rural” in another, due to small cell counts. Regional household median income was extracted from the 2016 Canadian Census data based on school postal codes; schools were dichotomized as being in the higher (>CAD 65,000) or lower (≤CAD 65,000) half of the sample based on the sample median. School delivery mode was defined as follows: 100% in-person, 100% online, mixed (students could attend in-person or online), blended (students alternated between in-person/online), or other.

#### 2.2.2. Physical Activity Programs

Interschool or Varsity: To measure whether schools continued to offer interschool or varsity sports, a question on the SPP asked schools to indicate whether interschool or varsity programs were running during the year, and response options included: “Yes, all varsity programs are running”, “Some programs are running, some have been cancelled”, and “No, all varsity programs have been cancelled”. The responses “Yes, all varsity programs are running” and “Some programs are running, some have been cancelled” were grouped due to small cell counts.

Intramural or Club: To measure whether schools continued to offer interschool or club activities and sports, a question on the SPP asked schools to indicate whether intramural programs/club activities were running during the year, and response options included: “Yes, all intramural programs are running”, “Some programs are running, some have been cancelled”, and “No, all intramural programs have been cancelled”. The responses “Yes, all intramural programs are running” and “Some programs are running, some have been cancelled” were grouped due to small cell counts.

Other Programs that Promote Physical Activity: PA programs not covered by previous questions were measured by asking schools whether any programs other than intramural and varsity sports were offered, with response options including: “Yes” or “No”. If schools had selected “Yes”, they were asked to indicate if the programs that were being offered were new to that school year. Response options included: “All programs are new this year”, “All programs are continuing from past years”, and “We have both new and continuing programs”. If new or continuing programs were being offered, two follow-up questions were asked: “what were the new programs?” and “what were the continuing programs?” allowing for open-ended text responses. 

#### 2.2.3. Physical Activity Staff Training and Facilities

Physical Activity Training: School staff training related to PA was measured by asking: “During the past 12 months, have school staff received training in the following areas: Physical activity (Check all that apply)”, with response options including: “In-service training (e.g., by Public/Regional Health)”, “Conferences, workshops or presentations”, “Teacher-initiated self-training on the internet”, and “Other”. Responses were then classified “Yes” or missing if the option was blank. The “Other” option was followed by a text-based response asking schools to describe what type of training was received by staff.

Changes to Facilities: To measure what changes schools made to PA facilities over the year, a question on the SPP asked: “During the past 12 months, were any changes made to physical activity facilities? (Check all that apply)”. Response options included: “New facilities were added”, “Old facilities were removed or closed”, and “Facilities were temporarily converted to classroom/study space”. For each option that was selected, a follow-up text-based question was asked to indicate what specific changes were made. Further, if changes were indicated, a question asked: “Were these facilities changes made as a result of COVID-19?”, with response options including either “Yes” or “No”.

### 2.3. Analysis

Descriptive analyses were conducted by cross-tabulating responses for PA programs, staff training, and facilities questions based on urbanicity, regional median income, and school delivery mode. Frequencies were compared based on urbanicity and regional median income categories using chi-square tests. Differences by school mode were not examined, as the expected frequencies of cross-tabulated cells were often less than 5.

Content analysis was conducted on open-ended text responses. Responses were generally sentence fragments or lists of programs being offered. French language responses (*n* = 29) were translated to English using Google Translate and edited by a fluent bilingual reader. Where possible, named programs without further description (e.g., “FillActive” or “iClimb”) were identified through secondary internet searches. The unit of analysis was set as each individual program. The beginning stages of the analysis included repeated reading and immersion within the data. All responses were read and analyzed individually by two members of the research team (H.C. and M.J.D). Open coding was performed by taking notes of initial impressions of the data. Further, reoccurring keywords were written and headings were produced to reflect the broader theme of the identified keywords, which produced the initial coding scheme. The initial codes were then further categorized based on their relationship to each other. Major and minor themes were then developed from the categories. After each researcher produced a list of major and minor themes independently, a second round of analysis included combining responses and further refining of themes and subthemes. Themes were discussed and fine-tuned until consensus was reached between both researchers. Responses that did not include sufficient information to be included in existing categories were removed from consideration. 

## 3. Results

### 3.1. Sample

Of the 132 schools that participated in the COMPASS study in 2020–2021, five schools did not submit the survey. Schools that did not submit the survey were in small urban or rural areas (*n* = 5), in lower-income regions (*n* = 4, higher income = 1), and predominantly delivered school content online (*n* = 3, blended = 1, other = 1). The 127 schools that submitted responses to the survey are described in Table 1, which cross-tabulates responses based on urbanicity, regional median income, and school delivery mode. Responding schools were in the provinces of Alberta (*n* = 5), British Columbia (*n* = 14), Ontario (*n* = 49), and Quebec (*n* = 59).

### 3.2. Interschool or Varsity Sport

Fourteen (11.1%) schools offered a varsity/interschool sports program during the school year, while 112 (88.9%) schools canceled all such programs. No significant differences were detected when comparing large/medium urban schools against small urban/rural schools χ^2^ (1, N = 126) = 1.77, *p* = 0.18; nor based on regional median income χ^2^ (1, N = 126) = 0.40, *p* = 0.53.

### 3.3. Intramural or Clubs

Forty-three (34.1%) schools offered at least some intramural programs or PA clubs during the school year, while 83 (65.9%) schools canceled their PA intramurals/clubs. No significant differences were detected when comparing schools in large/medium urban areas against schools in small urban/rural areas χ^2^ (1, N = 126) = 0.92, *p* = 0.33; nor based on regional median income χ^2^ (1, N = 126) = 2.08, *p* = 0.15.

### 3.4. PA Training

Staff completed some form of PA-related training at 90 (70.9%) schools. Training was predominantly in the form of conferences/presentations/workshops (*n* = 41) or self-initiated training through the internet (*n* = 35). Staff at large/medium urban schools were no more likely to complete training than staff at schools in small urban/rural areas χ^2^ (1, N = 127) = 0.002, *p* = 0.97; nor were there significant differences based on regional median income χ^2^ (1, N = 127) = 0.59, *p* = 0.44.

### 3.5. Other Programs to Promote Physical Activity

In total, 16 (12.6%) schools reported adding new PA-promoting programs for the 2020–2021 school year. Forty-nine (38.6%) schools reported continuing PA programs, of which 12 (24.5%) also reported new programs being implemented. Large/medium urban areas were no more likely to offer new programs than small urban/rural areas χ^2^ (1, N = 128) = 2.96, *p* = 0.09; nor were there significant differences based on regional median income χ^2^ (1, N = 128) = 0.45, *p* = 0.50.

Qualitative data to describe new programs was available for 16 schools. One entry was excluded from the analysis for describing a program that was claimed to be “not necessarily new”. Content analysis of new programs resulted in two major categories: “face-to-face” and “virtual” programs. These categories were both composed of two minor themes: “fitness and sport” and “general physical activity”. Of the new face-to-face programs, nine were fitness- and sport-oriented, and three were PA-oriented. Face-to-face fitness and sports programs that were introduced were organized sports such as basketball, general fitness, skill development, and strength training. Face-to-face PA programs mainly consisted of unstructured outdoor recreational activities (e.g., “teacher directed student breaks during the day where students walk the track” and “physical education classes have walked to golf ranges, tennis courts, hiking trails”). Of the new virtual programs, three were fitness- and sport-oriented, and two were PA-oriented. Virtual fitness and sports programs included virtual fitness and training clubs, and online fitness challenges offered to students. The two virtual PA programs included a response that described “general activities to get students moving”, and an online forum that was hosted by students to promote health.

Qualitative data to describe continuing programs were available for 49 schools, often listing multiple programs. Across the 49 schools, there were 76 reported continuing programs. Three entries were excluded for not providing enough information to judge the content due to only providing an acronym (“CLCC”), an ambiguous single word (“gym”), or explaining that the programs had been adapted to virtual delivery without describing the programs. Furthermore, four responses were excluded as their response indicated the program was canceled or on pause due to the COVID-19 pandemic. These programs included: “weight-room club and other fitness programs”, “dance club”, “student sport and student life”, and “outdoors”.

Content analysis of continuing programs resulted in five categories: sports programs (46), facilities and equipment access (13), PA- and health-promotion programs (8), clubs (5), and events (4). Continuing sports programs included multiple responses of basketball, hockey, dance, soccer, volleyball, badminton, and races or running programs. It also included programs that focused on sport and fitness development (e.g., “fitness fundamentals class”, “personal fitness class”, “sports performance classes” and “elite athlete class”), as well as any classes that involved sport or fitness education (e.g., “sport-studies program”, “sports-arts-studies”, and “fitness and performance class”). Facilities and equipment access included many responses indicating access to a weight room and/or cardio equipment (cardio bikes), as well as access to the gymnasium (e.g., “activities at the gym”, and “gym access for high needs students”). The PA- and health-promotion category was pertinent to programs that did not focus on fitness or skill development, but rather the promotion of PA through recreation or unstructured activities. This category was further refined into general physical education classes and outdoor programs, of which both included recreational activities (e.g., “personal health classes”, “classroom workshops”, and “outdoor education that promotes physical education not related to sport”). Continuing clubs included any type of explicit mention of a club that promoted PA, and it included responses indicating various running clubs, as well as a weight training club. Finally, the event themes included one-off events or occasions that promoted PA, and responses included events such as a health week, an outdoor/sports day, an outdoor race event, and Quebec-based FillActive events, which are non-competitive team sports and activities for girls.

### 3.6. Changes to Facilities

Forty-eight (37.8%) schools made some changes to their PA facilities. Of those 48, the majority (*n* = 30; 62.5%) indicated that the changes were made due to COVID-19. For schools that closed facilities (*n* = 23; 18.1%) or temporarily converted PA facilities to classrooms/study spaces (*n* = 14; 11%), the majority also indicated that the facility changes were due to COVID-19 (*n* = 20 and 12, respectively). Of the 20 (15.7%) schools that added new facilities, a quarter (*n* = 5) indicated changes were due to COVID-19. As multiple responses were permitted, two schools indicated they had simultaneously added new, closed old, and temporarily converted PA facilities. Two additional schools both introduced new facilities and closed old facilities, while three schools introduced new facilities and temporarily converted PA facilities. No schools indicated only closing old facilities and temporarily converting PA facilities into classrooms/study spaces without opening new facilities. 

Large/medium urban schools were at greater odds of having made any change to their PA facilities compared to schools in small urban/rural areas (odds ratio [95% confidence interval]: 2.3 [1.1, 4.8], χ^2^(1, N = 127) = 4.7, *p* = 0.03). However, among schools that did make a change, there were no differences based on urbanicity on the proportion that added new facilities, removed facilities, or converted PA facilities to classroom/study spaces, χ^2^(1, N = 48) = 0.04–0.8, *p* = 0.37–0.84. Additionally, large/medium urban schools were no more likely to have made changes as a result of COVID-19 compared to small urban/rural schools χ^2^(1, N = 127) = 4.733, *p* = 0.09.

The odds of having made any change to their PA facilities or having made a change due to COVID-19 did not differ based on regional median income; χ^2^(1, N = 127) = 0.03, *p* = 0.87 and χ^2^(1, N = 127) = 2.03, *p* = 0.36, respectively.

## 4. Discussion

This study investigated the changes Canadian secondary schools made to PA programs and facilities during the 2020–2021 academic year, the first full school year after the onset of the COVID-19 pandemic. As is consistent with the literature [21,22], the majority of schools in this sample reported canceling all interschool/varsity sports, intramural sports/clubs related to PA, and other programs to promote PA (in descending order of proportion). We found no statistically significant differences based on urbanicity nor regional median income in the odds of schools offering interschool or varsity sports, intramural programs or clubs to students, or staff training related to PA. Schools in large/medium urban areas were more likely to have made a change to their PA facilities compared to small urban/rural areas, but there were no differences in the types of facility changes between regions. Unsurprisingly, in schools that reported making some changes to their PA facilities, the majority indicated that these changes were due to the pandemic. Nonetheless, a surprising majority of schools reported that staff had completed PA-related training, and there was also evidence of resilience and resourcefulness in many schools that reported new and adapted PA programs and facilities, particularly related to sports and fitness. 

Previous research has found widespread restrictions and the prohibition of structured PA opportunities in schools during the initial lockdown stages of the pandemic [23,24,25]. As a result of school closures hindering indoor PA facilities and programs, studies have consistently reported lower levels of PA among adolescents, with little or no significant increases in outdoor PA activities to compensate, as a result of government-imposed restrictions [26,27]. While our results also support widespread cancelations in PA programs during the ongoing pandemic response, as expected due to the ongoing restrictions [23,24,25], they also highlight several additions and continuations of PA programs by schools. Approximately 44% of participating schools reported maintaining or introducing programs in the 2020–2021 school year. Sixteen schools introduced 17 novel face-to-face and virtual PA programs, which consisted of both programs to promote general PA, as well as programs oriented specifically towards fitness or sports, though the latter was more common. Forty-nine of the 127 schools continued offering 76 programs, which consisted of sports programs, programs to access PA facilities and equipment, and programs to promote PA and health, clubs, and events. Schools demonstrated innovation and resourcefulness in being able to implement new programs despite the drastic limitations set by the COVID-19 pandemic. Further, the effort placed into continuing programs throughout the restrictions should be celebrated, and highlighted PA-promotion techniques that are robust to disruptions.

The qualitative responses from this survey provide some insight into the types of PA programs that schools attempted to implement (face-to-face and virtual fitness and physical activity programs) and maintain (sports programs, facilities access, PA and health promotion, clubs, and events) despite COVID-19-related restrictions. Previous research determined that students participated less in virtual PA when compared to face-to-face instruction because of a lack of adequate facilities, equipment, and technology [28]. Even when in-person facilities were available, students and families expressed concern about exposure to COVID-19, which limited their ability to access these physical activities [28]. The classifications identified for both newly implemented and continuing programs may help future research to examine the efficacy of programs in terms of maintaining PA levels among students under these circumstances, which is of particular importance as creating ineffective programs may needlessly put stress on school staff to facilitate these programs [29].

The findings from the current study demonstrate the number of canceled programs and opportunities for PA that schools provide outside of physical education classes. Though physical education is essential in providing children with physical literacy, motivation, knowledge, and skills for PA [30], it may not be sufficient in ensuring adolescents meet PA guidelines; identifying programs that are robust against closures and restrictions may offer insight into opportunities for PA [31]. Additionally, we found that despite the challenges posed by the COVID-19 pandemic, a majority of the schools in our sample reported staff having completed PA-related training, which is crucial in school-based PA program implementation and overall physical education outcomes [14,15]. Training was primarily via conferences, workshops, and presentations, yet this was followed by staff-initiated training online, highlighting the commitment of school staff/educators to supporting student health. Future research should further explore the nature and content of this training to provide practical insights into the needs of the school staff. As suggested by extant literature [14], such training opportunities may have helped schools in the delivery of physical education and PA programming during the ongoing pandemic response and facilitated a return to typical school delivery after COVID-19 pandemic protocols were lifted. Longitudinal studies examining whether staff training is associated with school PA delivery changes and student engagement in school-based PA opportunities may prove valuable. 

Longitudinal research is also needed to examine whether cut programs or facilities are restored to at least pre-pandemic levels, and if new and adapted programs and facilities endure. On a positive note, there were few differences in PA resources between regions based on either urbanicity or regional median income. Given concerns that health inequities have been exacerbated due to the emergence of COVID-19 [32,33,34], a positive finding was that in this sample of Canadian secondary schools, schools located in lower-income regions were at no greater odds of losing PA programs or falling behind on the creation of new PA facilities compared to schools in higher-income regions. Similarly, while regions differed on facility changes based on urbanicity, there were no disparities in the introduction of new facilities, closure of old facilities, or conversion of existing facilities to classroom/study spaces. However, any pre-existing disparities in PA programs and facilities from before the pandemic likely still exist [12,13], as there were no improvements by socioeconomic or urbanicity status. Therefore, it is recommended that a sustained effort is made in the surveillance of school PA programs and policies to ensure recovery to levels even higher than those observed prior to the pandemic. Furthermore, continued surveillance may allow for PA programs to be easily adapted to formats that are more desirable and optimize student participation, particularly in the event of future disruptions. There is a need to address key barriers to PA, including a lack of time, motivation, accessible places, and school resources [35,36]. Longitudinal studies are warranted on the resilience of programs and the persistence of newly added ones.

Several limitations to the study must be considered. First, data were used from a convenience sample of schools participating in an annual survey: the sample is inherently biased to the 127 schools which are located within four of Canada’s thirteen provinces and territories and may not reflect what changes schools made in other parts of the country, particularly given differences in COVID-19 policies. It is also plausible that schools that agreed to participate in the study differ from other schools, such as in resources and prioritization of student health. Furthermore, while schools are provided their responses from previous years of participation to assist in recall and in cases of new school administrators, it is plausible that school respondents may have forgotten or been unaware of various new programs (e.g., virtual clubs, walking breaks), particularly during this period of disruption and competing priorities. Also, schools were encouraged to complete the survey as a team to help mitigate inaccurate responses due to a lack of awareness, but doing so may have been less feasible during the pandemic response. While confidential, school reports may have also been influenced by social desirability bias, given the pressures placed on schools to support student health and well-being. Additionally, many of the open-ended responses were lists of sports, especially among continuing programs, and it was unclear whether these sports represented something other than varsity or intramural sports participation. Similarly, some open-ended descriptions of new programs represent modifications to existing PA programming (e.g., “physical education classes have walked to golf ranges, tennis courts, hiking trails”) rather than being genuinely “new”. Measures regarding staff training were limited to the delivery and source of the training and did not provide important details on the nature and content of the training programs. It should also be noted that nearly 40% of the sample was schools that continued to deliver classes entirely online, and another 28% of schools were delivering some content online in either a blended or mixed system. Given the high rates of online delivery, it is unsurprising that many programs were cancelled for the year. Due to statistical limitations (a high number of categories leading to low frequencies in cells), no testing was performed to compare survey responses based on school mode; however, the proportion of cancelled programs unsurprisingly appears higher for online delivery modes than in-person. Chi-square tests were used as opposed to Fischer’s exact tests, as they are generally considered appropriate when no more than 20% of the expected cell counts are less than five and individual expected cell counts are at least one. Nonetheless, the low frequencies in some categories may have still limited the power of analyses examining differences between urbanicity and regional median income in the various PA program and facility changes.

## 5. Conclusions

This study serves as a reminder for stakeholders of the number of school-related PA programs that were affected by the COVID-19 pandemic even as schools proceeded with a full academic year. Now that students have fully returned to in-person learning, it is important that schools recommence programs to support a return to PA among adolescents. Given the high rates of PA and sport cancelation at the intramural level or above, potential impacts on the fitness and physical literacy levels of students may represent a substantial barrier to participation in their future. Modifications and new programming opportunities implemented by schools may offer ways of reaching students who may be less likely to participate in PA in typical school contexts. Longitudinal research is needed regarding the pandemic recovery phase to examine whether programs returned to pre-pandemic levels or whether changes were sustained. It is important to ensure that there are no disparities in schools’ abilities to return to pre-pandemic PA program delivery levels. 

## Figures and Tables

**Table 1 ijerph-21-00788-t001:** Survey response frequency for the full sample and by urbanicity, regional median income, and school delivery mode.

	All Responses	Urbanicity		Regional Median Income		School Delivery Mode				
		Large/Medium Urban	Small Urban/Rural	Lower (≤CAD 65,000)	Higher (>CAD 65,000)	In- Person	Online	Mixed	Blended	Other
Total	127	69	58	62	65	16	50	13	23	25
Interschool or varsity programs										
Yes	14 (11.1)	10 (14.5)	4 (7)	8 (12.9)	6 (9.4)	2 (12.5)	2 (4.1)	0 (0)	5 (21.7)	5 (20)
All Canceled	112 (88.9)	59 (85.5)	53 (93)	54 (87.1)	58 (90.6)	14 (87.5)	47 (95.9)	13 (100)	18 (78.3)	20 (80)
Missing/blank	1	0	1	0	1	0	1	0	0	0
Intramural/ club activities										
Yes	43 (34.1)	21 (30.4)	22 (38.6)	25 (40.3)	18 (28.1)	6 (37.5)	14 (28.6)	3 (23.1)	11 (47.8)	9 (36)
All Canceled	83 (65.9)	48 (69.6)	35 (61.4)	37 (59.7)	46 (71.9)	10 (62.5)	35 (71.4)	10 (76.9)	12 (52.2)	16 (64)
Missing/blank	1	0	1	0	1	0	1	0	0	0
Other Programs to Promote PA										
Yes	56 (44.1)	35 (50.7)	21 (36.2)	29 (46.8)	27 (41.5)	5 (31.3)	20 (40)	4 (30.8)	14 (60.9)	13 (52)
New *	17 (13.4)	15 (21.7)	2 (3.4)	6 (9.7)	11 (16.9)	1 (6.3)	6 (12)	1 (7.7)	4 (17.4)	5 (20)
Continuing *	51 (40.2)	31 (44.9)	20 (34.5)	27 (43.5)	24 (36.9)	5 (31.3)	18 (36)	4 (30.8)	13 (56.3)	11 (44)
No	71 (55.9)	34 (49.3)	38 (65.5)	33 (53.2)	38 (58.5)	11 (68.8)	30 (60)	9 (69.2)	9 (39.1)	12 (48)
PA Training										
Any	72 (56.7)	39 (56.5)	33 (56.9)	33 (53.2)	39 (60)	8 (50)	28 (56)	8 (61.5)	16 (69.6)	12 (48)
In-service training *	11 (8.7)	6 (8.7)	5 (8.6)	8 (12.9)	3 (4.6)	2 (12.5)	6 (12)	0 (0)	1 (4.3)	2 (8)
Conferences, workshops, presentations *	41 (32.3)	23 (33.3)	18 (31)	18 (29)	23 (35.4)	5 (31.3)	19 (38)	4 (30.8)	7 (30.4)	6 (24)
Teacher initiated self-training through the internet *	35 (27.6)	18 (26.1)	17 (29.3)	15 (24.2)	20 (30.8)	5 (31.3)	11 (22)	5 (38.5)	10 (43.5)	4 (16)
Other *	3 (2.4)	2 (2.9)	1 (1.7)	2 (3.2)	1 (1.5)	1 (6.3)	0 (0)	0 (0)	2 (8.7)	0 (0)
None Indicated	55 (43.3)	30 (43.5)	26 (44.8)	29 (46.8)	27 (41.5)	8 (50)	22 (44)	5 (38.5)	7 (30.4)	13 (52)
PA Facility Changes										
Any	48 (37.8)	32 (46.4)	16 (27.6)	23 (37.1)	25 (38.5)	9 (56.3)	18 (36)	3 (23.1)	8 (34.8)	10 (40)
New facilities added *	20 (15.7)	13 (18.8)	7 (12.1)	10 (16.1)	10 (15.4)	3 (18.8)	8 (16)	0 (0)	4 (17.4)	5 (20)
Old facilities closed/removed *	23 (18.1)	14 (20.3)	9 (15.5)	9 (14.5)	14 (21.5)	5 (31.3)	8 (16)	2 (15.4)	3 (13)	5 (20)
Facilities temporarily converted to learning space *	14 (11)	8 (11.6)	6 (10.3)	7 (11.3)	7 (10.8)	3 (18.8)	5 (10)	1 (7.7)	1 (4.3)	4 (16)
None Indicated	79 (62.2)	37 (53.6)	42 (72.4)	39 (62.9)	40 (61.5)	7 (43.8)	32 (64)	10 (76.9)	15 (65.2)	15 (60)
Facility Changes due to COVID-19										
Yes	30 (62.5)	20 (29)	10 (17.2)	12 (19.4)	18 (27.7)	6 (37.5)	11 (22)	3 (23.1)	4 (17.4)	6 (24)
No	18 (37.5)	12 (17.4)	6 (10.3)	11 (17.7)	7 (10.8)	3 (18.8)	7 (14)	0 (0)	4 (17.4)	4 (16)

Note: Of 132 schools participating in the COMPASS study, five did not submit survey responses. * Indicates responses that are not mutually exclusive. Except for the total row, percentages are calculated within column strata and do not account for non-submitted surveys or missing responses. Mixed school mode meant students could attend in-person or online; hybrid school mode meant students alternated between in-person and online.

## Data Availability

COMPASS study data is available upon request through completion and approval of an online form: https://uwaterloo.ca/compass-system/information-researchers/data-usage-application. The datasets used during the current study are available from the corresponding author on reasonable request.

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
