# Peer review of "Changes to Secondary School Physical Activity Programs and Policy after Emerging from COVID-19 Lockdowns"

_ijerph, 2024, doi:10.3390/ijerph21060788_

Round 1

Reviewer 1 Report

Comments and Suggestions for Authors

Karen A. Patte et al. submitted to IJERPH an article, dealing with the changes to secondary school physical activity programs and policy after emerging from pandemic lockdowns in Canada.

This study investigated the changes in Canadian secondary schools made to physical activity programs and facilities during the 2020-2021 academic year, practically 4 years ago.

In fact, the study appears dated and obsolete and no longer in line with current reality.

It should probably be implemented by carrying out a follow-up evaluation, to possibly understand the changes over time or verify the consolidation of the changes occurred.

Comments on the Quality of English Language

Moderate editing of English language required

Author Response

COMMENT 1: This study investigated the changes in Canadian secondary schools made to physical activity programs and facilities during the 2020-2021 academic year, practically 4 years ago. In fact, the study appears dated and obsolete and no longer in line with current reality. It should probably be implemented by carrying out a follow-up evaluation, to possibly understand the changes over time or verify the consolidation of the changes occurred.

Our Response: While we were initially disheartened by this comment, we appreciate the push to further consider the value of this current study, and to clarify and emphasize this in the revised manuscript. We purposefully used data from the first full school year following the onset of the COVID-19 pandemic. The pandemic onset was in March 2020, and we used data from the 2020-2021 school year. During this year, there were partial school closures, different learning models implemented, and other policies and practices. We chose this year to examine the changes that schools made in their physical activity programs and facilities as a time when there were disruptions, to examine how schools were impacted and how they adapted to these unprecedented challenges. While much has been speculated about such changes, actual documented evidence is scarce and important, particularly for policy and decision makers. Our study actually shows a more nuanced picture of school PA context (e.g., with many continued programs, and new or adapted programs and facilities), than may be presumed. There have also now been several studies documenting declines in adolescents physical activity levels and speculation that is these declines reflect cancellations and closures, but again, scarce studies that have actually assessed the physical activity environment. By examining how schools were impacted and adapted, we believe this evidence is valuable to inform practical recommendations for schools in future cases of disruptions, closures, or other unprecedented circumstances. The findings of this study might provide insight as to what school physical activity programs are more robust and adaptable to promote adolescent physical levels during periods of significant challenges to young people’s physical activity. Results also provide interesting and valuable examples of how schools were resourceful in adapting and adding new programs, including online offerings, which point to potential novel ways of promoting physical activity among adolescents. We have revised the manuscript (e.g., by adding to the introduction and revising the abstract) to help to clarify and emphasize the purpose and ongoing value of this study. We agree that follow up evaluation is worthwhile, and we have added further mention and direction for this purpose.

Reviewer 2 Report

Comments and Suggestions for Authors

Dear authors.

Congratulations on your work. It is certainly always a merit to conclude an investigation like this.

Apart from the personal opinion that a proposal of this type may generate, in relation to the greater or lesser contribution to the field of Physical Education, it is necessary to make a reference to improvement and I think it is necessary to pay attention to it.

Specifically, I am concerned about the evaluation instrument used (SPP). This is a modification of the Healthy School Planner tool, about which we know nothing. What is more, on entering the website provided by the authors, we find that it is completely disabled and that it is not possible to obtain relevant information about it. In this sense, it is necessary to provide information on the psychometric properties of the instrument used. On the other hand, mention is made of aspects such as, for example, the inclusion of questions related to substance use, something that has little to do with the aim of the study and which is not really apparent in the results. It is not consistent. Moreover, if you go to the project's website, there is no such relevant information either.

I think this is sufficiently important to be corrected before the publication of the work.

Best regards.

Author Response

Reviewer 2

COMMENT 1: Specifically, I am concerned about the evaluation instrument used (SPP). This is a modification of the Healthy School Planner tool, about which we know nothing. What is more, on entering the website provided by the authors, we find that it is completely disabled and that it is not possible to obtain relevant information about it. In this sense, it is necessary to provide information on the psychometric properties of the instrument used.

Our Response: Thank you for notifying us that the link for the Healthy School Planner (HSP) toll is disabled. The Pan-Canadian Joint Consortium for School Health (JCSH) has discontinued the HSP program. We have amended the previous citation with an updated citation that provides background information on the tool.

We have also added further information on the specific SPP measures used in the current study, including citation to a technical report on the SPP development (See page 3, lines 101-119).

COMMENT 2: On the other hand, mention is made of aspects such as, for example, the inclusion of questions related to substance use, something that has little to do with the aim of the study and which is not really apparent in the results. It is not consistent. Moreover, if you go to the project's website, there is no such relevant information either.

Our Response: While the survey does include questions in areas such as substance use, we agree that it is not relevant to our study, and as such, any mention of has been removed in the revised manuscript. We have focused the description of the methods on the measures used in the current study.

Reviewer 3 Report

Comments and Suggestions for Authors

Dear Authors,

First of all, I would like to congratulate you on the completion of this research. It is interesting and seems well developed. However, the manuscript presents formal errors and methodological limitations that should be addressed before its possible publication in this Journal.

ABSTRACT:
Abbreviations in this section are discouraged. Please remove them.
The abstract should include quantitative results (not just their qualitative transcription or interpretation).

INTRODUCTION:
Both this section and the Discussion obviate key references on this topic (DOI: 10.3390/ejihpe11030076).

This section should be slightly lengthened by adequately contextualizing the object of study of the research.

METHODS/RESULTS:
It should be complemented with the calculation of the sample size needed to achieve the desired confidence margin.
Zeros as the last decimal place do not mean anything. They should be eliminated.
The effect sizes of the final sample analyzed and of the statistical techniques used should be included.

DISCUSSION:
The quantity and quality of the bibliographic references used should be increased. There are statements that are not referenced in any way.

Kind regards

Author Response

Reviewer 3

COMMENT 1: ABSTRACT: Abbreviations in this section are discouraged. Please remove them.

Our Response: We have removed the physical activity (PA) abbreviation from the Abstract.

COMMENT 2: The abstract should include quantitative results (not just their qualitative transcription or interpretation).

Our Response: We have considerably revised the abstract, with a focus on the quantitative results, and reduced mention of the content analysis results.

COMMENT 3: INTRODUCTION: Both this section and the Discussion obviate key references on this topic (DOI: 10.3390/ejihpe11030076). This section should be slightly lengthened by adequately contextualizing the object of study of the research.

Our Response: We appreciate the suggestion for a reference, but have opted not to include it because we had difficulty understanding the relevance of the suggested study on academic experiences in postsecondary to our current study of physical activity programs and facilities in high schools. We welcome further suggestions of key references that more directly relate to our study. We have added to the introduction, including a new citation, to help provide further context to the study and our objectives (Page 2, lines 81-90).

COMMENT 4: METHODS/RESULTS:  It should be complemented with the calculation of the sample size needed to achieve the desired confidence margin. Zeros as the last decimal place do not mean anything. They should be eliminated. The effect sizes of the final sample analyzed and of the statistical techniques used should be included.

Our Response: We included zeros as the last decimal place for consistency, but have removed any zeros as the last decimal place as advised. Our study was primarily descriptive, including frequencies and qualitative content analysis.

Also, there is debate about the value of post hoc power analyses – for example:

  • Althouse, A. (2021). Post hoc power: Not empowering, just misleading. Journal of Surgical Research, 259, A3--A6.
  • Hoenig, J. M., & Heisey, D. M. (2001). The abuse of power: The pervasive fallacy of power calculations for data analysis. The American Statistician, 55, 19--24.

However, for the Chi-square analyses of differences by school area urbanicity and income we have added to our limitations section: “Chi-square tests were used as opposed to Fischer’s Exact test, as they are generally considered appropriate when no more than 20% of the expected cell counts are less than five and individual expected cell counts are at least one. Nonetheless, the low frequencies in some categories may have still limited the power of analyses examining differences between urbanicity and regional median income in the various PA program and facility changes.”

COMMENT 5: DISCUSSION: The quantity and quality of the bibliographic references used should be increased. There are statements that are not referenced in any way.

Our Response: We have carefully reviewed our Discussion section to ensure that statements that require citations do have them. Sentences that remain without citations are summaries and discussion of our findings, suggestions for future research, etc.

Reviewer 4 Report

Comments and Suggestions for Authors

Dear Authors,

I hope this message finds you well. I have had the pleasure of reviewing your manuscript, "Changes to Secondary School Physical Activity Programs and Policy After Emerging from COVID-19 Lockdowns," and I appreciate the significant effort you have put into addressing this important and timely topic. I am writing to provide detailed feedback on your study, with a focus on constructiveness and openness to dialogue for any suggested changes or improvements.

Your manuscript presents a comprehensive analysis of the adaptations made by Canadian secondary schools to their physical activity (PA) programs and facilities during the first full school year following the COVID-19 pandemic. The exploration of disparities based on urbanicity and income, combined with both quantitative and qualitative analyses, adds substantial value to the existing literature on this subject. This work is particularly relevant given the widespread impact of the pandemic on adolescent health behaviors.

The use of the COMPASS study framework to collect annual survey data from a convenience sample of Canadian secondary schools is a robust approach for longitudinal analysis. This design allows for a detailed examination of how schools adapted their PA programs in response to the pandemic. However, I suggest explicitly acknowledging the limitation of using a convenience sample, as this may affect the generalizability of your findings. Including a brief discussion of this limitation in your discussion section would provide a more balanced perspective on the study's findings.

The adaptation of the School Programs and Policies Questionnaire (SPP) from the Healthy School Planner tool is appropriate and demonstrates a thoughtful approach to capturing relevant data. The mixed-methods design, incorporating both qualitative and quantitative items, is a significant strength of your study. This approach allows for a richer understanding of the changes in PA programs. Nonetheless, the reliance on self-reported data could introduce bias, and it would be beneficial to highlight the measures you took to mitigate this potential bias, such as providing school contacts with their previous years’ survey responses to assist with recall.

Your descriptive analyses, using cross-tabulations and chi-square tests, effectively examine differences based on urbanicity and regional median income. This analytical approach is suitable for identifying significant patterns and trends. However, the low frequency in some categories may affect the reliability of these tests. Discussing this limitation and considering alternative statistical methods, where applicable, would enhance the robustness of your findings.

The qualitative analysis using NVivo software is well-executed. Categorizing responses into major and minor themes provides clarity and depth to the data interpretation. To further enhance transparency and replicability, consider providing more details on the coding process and measures of inter-coder reliability. This additional information would strengthen the methodological rigor of your qualitative analysis.

One of the key strengths of your study is the detailed examination of school staff training related to PA. While you discuss the types of training received, offering more detailed information on the nature and content of these training programs would be valuable. Including specific examples or case studies of training initiatives could provide practical insights and enhance the applicability of your findings.

The discussion of limitations in your study is thorough, but it could be strengthened by explicitly addressing the sample selection, self-reported data, and statistical power. Providing a comprehensive overview of these limitations would offer a more nuanced understanding of the study’s constraints and contextualize your findings more effectively.

Your manuscript also benefits from a thoughtful exploration of future research directions. Expanding this section to include suggestions for longitudinal studies that track the long-term impact of COVID-19 on PA programs or recommendations for implementing resilient PA programs in schools would be highly beneficial. This forward-looking perspective would underscore the ongoing relevance of your research.

In conclusion, your study offers important insights into the changes and challenges faced by secondary schools in promoting physical activity during the pandemic. The methodology is sound, and the findings are highly relevant to the current educational and public health landscape. By addressing the points highlighted above, you can further enhance the clarity and impact of your work.

I appreciate the opportunity to review your manuscript and hope that my comments are helpful. I am open to further discussion and collaboration to support the refinement of your study. Your research is a valuable contribution to understanding how schools can adapt to unprecedented challenges and continue to promote the health and well-being of their students.

Best regards,

Comments on the Quality of English Language

English language and style are fine/minor spell check required

Author Response

Reviewer 4

COMMENT 1: The use of the COMPASS study framework to collect annual survey data from a convenience sample of Canadian secondary schools is a robust approach for longitudinal analysis. This design allows for a detailed examination of how schools adapted their PA programs in response to the pandemic. However, I suggest explicitly acknowledging the limitation of using a convenience sample, as this may affect the generalizability of your findings. Including a brief discussion of this limitation in your discussion section would provide a more balanced perspective on the study's findings.

Our Response: Thank you taking the time to review our paper and provide thoughtful suggestions. We appreciate the positive feedback. We had already included the usage of a convenience sample as a limitation of the study, but we have expanded on this limitation in the revised manuscript (Page 10, lines 389-394): “First, data were used from a convenience sample of schools participating in an annual survey: the sample is inherently biased to the 127 schools which are located within four of Canada’s thirteen provinces and territories and may not reflect what changes schools made in other parts of the country, particularly given differences in COVID-19 policies. It is also plausible that schools that agreed to participate in the study differ from other schools, such as in resources and prioritization of student health”

COMMENT 2: The adaptation of the School Programs and Policies Questionnaire (SPP) from the Healthy School Planner tool is appropriate and demonstrates a thoughtful approach to capturing relevant data. The mixed-methods design, incorporating both qualitative and quantitative items, is a significant strength of your study. This approach allows for a richer understanding of the changes in PA programs. Nonetheless, the reliance on self-reported data could introduce bias, and it would be beneficial to highlight the measures you took to mitigate this potential bias, such as providing school contacts with their previous years’ survey responses to assist with recall.

Our Response: Thank you for your positive feedback regarding the methodology of the study. We have expanded our methods section on the description of the SPP tool and it’s development and administration (lines 101-119), including the addition of a citation to a technical report with further information on it’s development and implementation.

We have also added to our limitations section in our discussion of potential bias (lines 397-403). For example: “While schools are provided their responses from previous years of participation to assist in recall and in cases of new school administrators, it is plausible that school respondents may have forgotten or been unaware of various new programs (e.g., virtual clubs, walking breaks), particularly during this period of disruption and competing priorities. Also, schools are encouraged to complete the survey as a team, to help mitigate inaccurate responses due to a lack of awareness, but doing so may have been less feasible during the pandemic response. While confidential, school reports may have also been influenced by social desirability bias, given the pressures placed on schools to support student health and well-being.”

COMMENT 3: Your descriptive analyses, using cross-tabulations and chi-square tests, effectively examine differences based on urbanicity and regional median income. This analytical approach is suitable for identifying significant patterns and trends. However, the low frequency in some categories may affect the reliability of these tests. Discussing this limitation and considering alternative statistical methods, where applicable, would enhance the robustness of your findings.

Our Response: Thank you for your feedback surrounding the analytical approach used in the study. We have added to the limitations section regarding the low frequency in some categories (line 415-420): “Chi-square tests were used as opposed to Fischer’s Exact test, as they are generally considered appropriate when no more than 20% of the expected cell counts are less than five and individual expected cell counts are at least one. Nonetheless, the low frequencies in some categories may have still limited the power of analyses examining differences between urbanicity and regional median income in the various PA program and facility changes.”

COMMENT 4: The qualitative analysis using NVivo software is well-executed. Categorizing responses into major and minor themes provides clarity and depth to the data interpretation. To further enhance transparency and replicability, consider providing more details on the coding process and measures of inter-coder reliability. This additional information would strengthen the methodological rigor of your qualitative analysis.

Our Response: We are pleased to receive this positive feedback regarding the qualitative analysis. We have amended the analysis section of the methods to provide more detailed information on the coding process such that it can be followed clearly and step-by-step, increasing its replicability (lines 190-196). While we did not explicitly calculate inter-coder reliability, our two coders were remarkably consistent, with the list of themes/categories nearly identical.

COMMENT 5: One of the key strengths of your study is the detailed examination of school staff training related to PA. While you discuss the types of training received, offering more detailed information on the nature and content of these training programs would be valuable. Including specific examples or case studies of training initiatives could provide practical insights and enhance the applicability of your findings.

Our Response: We agree that more detailed information would be quite valuable on these training programs. Unfortunately, only the types of training received was available to us, and we do not have information on the nature and content of these training programs. Of the 3 schools that selected “other” for the type of training received, and opted to include an open-ended text response to describe this further, none of the responses indicated training programs. Therefore, aside from the binary responses to each of the training categories, it would be difficult to provide specific examples or case studies of training initiatives as this information is not available to us. We have added to our discussion on staff training, including suggesting further research (Lines 365-369: “…Training was primarily via conferences, workshops, and presentations, yet this was fol-lowed by staff-initiated training online, highlighting the commitment of school staff/educators to supporting student health. Future research should further explore the nature and content of this training to provide practical insights on the school staff needs.”

We have added mention of this limited information as a limitation of the study (page 10, lines 408-410): “Measures regarding staff training were limited to the delivery and source of the train-ing, and did not provide important details on the nature and content of training programs.”

COMMENT 6: The discussion of limitations in your study is thorough, but it could be strengthened by explicitly addressing the sample selection, self-reported data, and statistical power. Providing a comprehensive overview of these limitations would offer a more nuanced understanding of the study’s constraints and contextualize your findings more effectively.

Our Response: We have reviewed and expanded on our limitations section to ensure we have explicitly addressed these limitations to contextualize the study appropriately.

Several limitations to the study must be considered. First, data were used from a convenience sample of schools participating in an annual survey: the sample is inher-ently biased to the 127 schools which are located within four of Canada’s thirteen provinces and territories and may not reflect what changes schools made in other parts of the country, particularly given differences in COVID-19 policies. It is also plausible that schools that agreed to participate in the study differ from other schools, such as in resources and prioritization of student health. Furthermore, while schools are provided their responses from previous years of participation to assist in recall and in cases of new school administrators, it is plausible that school respondents may have forgotten or been unaware of various new programs (e.g., virtual clubs, walking breaks), particularly during this period of disruption and competing priorities. Also, schools are encouraged to complete the survey as a team, to help mitigate inaccurate responses due to a lack of awareness, but doing so may have been less feasible during the pandemic response. While confidential, school reports may have also been influenced by social desirability bias, given the pressures placed on schools to support student health and well-being. Additionally, many of the open-ended responses were lists of sports, especially among continuing programs, and it was unclear whether these sports represent something other than varsity or intramural sport participation. Similarly, some open-ended descriptions to new programs represent modifications to existing PA programming (e.g., “physical education classes have walked to golf ranges, tennis courts, hiking trails”) rather than being genuinely “new”. Measures regarding staff training were limited to the delivery and source of the training, and did not provide important details on the nature and content of training programs. It should also be noted that nearly 40% of the sample was schools who continued to deliver classes entirely online, and another 28% of schools were delivering some content online in either a blended or mixed system. Given the high rates of online delivery it is unsurprising that many programs were cancelled for the year. Due to statistical limitations (high number of categories leading to low frequencies in cells), no testing was performed to compare survey responses based on school mode, however the proportion of cancelled programs unsurprisingly appears higher for online delivery modes than in-person. Chi-square tests were used as opposed to Fischer’s Exact test, as they are generally considered appropriate when no more than 20% of the ex-pected cell counts are less than five and individual expected cell counts are at least one. Nonetheless, the low frequencies in some categories may have still limited the power of analyses examining differences between urbanicity and regional median income in the various PA program and facility changes.”

COMMENT 7: Your manuscript also benefits from a thoughtful exploration of future research directions. Expanding this section to include suggestions for longitudinal studies that track the long-term impact of COVID-19 on PA programs or recommendations for implementing resilient PA programs in schools would be highly beneficial. This forward-looking perspective would underscore the ongoing relevance of your research.

Our Response: We agree with the reviewer in the value of longitudinal studies for these purposes. As such, to expand on this discussion, we have added more future research recommendations, including specifying longitudinal studies.

For example, lines 372-377: “Longitudinal studies examining whether staff training is associated with school PA delivery changes and student engagement in school-based PA opportunities may prove valuable. Longitudinal research is also needed to examine whether cut programs or facilities are restored to at least pre-pandemic levels, and if new and adapted programs and facilities endure… Longitudinal studies are warranted on the resilience of programs and the persistence of newly added ones.”

Also in our conclusion on lines 372-3: “Longitudinal research is needed into the pandemic recovery phase, to examine whether programs returned to pre-pandemic levels or whether changes were sustained. It will be important to ensure that there were no disparities in schools’ ability to return to pre-pandemic PA program delivery.”

Round 2

Reviewer 1 Report

Comments and Suggestions for Authors

You will agree that this manuscript currently describes an overview that no longer fits the current situation, which absolutely needs to be investigated through a follow-up study. Thanks for the improvements made.

Comments on the Quality of English Language

Minor editing of English language required

Reviewer 3 Report

Comments and Suggestions for Authors

Dear Authors,

The Authors have done a great job of correcting and improving the text presented. Consequently, I now consider that it can be accepted for publication in this Journal.

Kind regards

Reviewer 4 Report

Comments and Suggestions for Authors

Dear authors, congratulations

Thank you for responding to all my recommendations in a constructive way.

For my part, the manuscript can be accepted in the current version.

Best regards